

# Real-Time Biological Early Warning System based on Freshwater Mussels' Valvometry Data

Ashkan Pilbala [1], Nicoletta Riccardi [2], Nina Benistati [3], Vanessa Modesto [2], Donatella Termini [3,4], Dario Manca [2], Augusto Benigni [5], Cristiano Corradini [5], Tommaso Lazzarin [6], Tommaso Moramarco [5], Luigi Fraccarollo [1], and Sebastiano Piccolroaz [1,*]

[1]University of Trento, Department of Civil, Environmental and Mechanical Engineering, Trento, Italy
[2]The National Research Council (CNR) – Water Research Institute (IRSA), Verbania, Italy
[3]University of Palermo, Department of Engineering, Palermo, Italy
[4]NBFC, National Biodiversity Future Center, Palermo, Italy
[5]The National Research Council (CNR) – Research Institute for Geo Hydrological Protection (IRPI), Perugia, Italy
[6]University of Padova, Department of Civil, Environmental and Architectural Engineering, Padova, Italy
[*]Corresponding author: s.piccolroaz@unitn.it

**Abstract.** The aim of this study is to investigate the impact of natural river floods on biotic communities. To this purpose, we used freshwater mussels (FMs), recognized as one of the most reliable bioindicators in aquatic environments. A well-established valvometry technique was applied to measure the FMs valve gaping behaviour, considering both gaping amplitude and frequency. The mussels have been employed in two distinct configurations, either free to move or stuck on vertical bars.

We performed experiments in a laboratory flume and in the Paglia River (Italy). The FMs valve gaping movement was first recorded, then the continuous wavelet transform (CWT) analysis was applied to the signals to get the time-dependent frequency of the signals. Laboratory experiments allowed to assess to what extent stuck mussels react differently than free mussels to abrupt increases in flow conditions. Subsequently, we examined the response of thirteen stuck mussels installed in real riverine conditions during a moderate flood occurred on March 31, 2022, with a rapid increase of the water level. The experimental

results demonstrate that stuck mussels produce signals that are more consistent and easier to interpret compared to free mussels, primarily due to the reduced number of features resulting from movement constraints. The stuck mussels in the field showed a sharp and timely change of valve gaping frequency as the flood ramped up, thus confirming the findings in the laboratory. The results highlight the effectiveness of using FMs as bioindicators for assessing the impact of floods on the aquatic ecosystem, and the utility of CWT as a suitable signal processing tool for analyzing valvometric time series. These findings provide a

pathway towards the integration of FMs valvometry and CWT for the development of operational real-time Biological Early Warning Systems (BEWS) aimed at the monitoring and safeguarding of aquatic ecosystems.



## 1 Introduction

Sustainable water resource management requires the protection of water-dependent ecosystems, as they play a pivotal role in
maintaining the ecological balance and overall health of our water resources (Zieritz et al., 2022; Makanda et al., 2022). This is
a challenging task, that is further compounded by the ongoing effects of climate change on water resources, which intensifies
conflicts related to water resource allocation. Indeed, besides impacting water availability and quality, climate change influences
water demand, thereby affecting the availability of water needed to sustain the ecological functioning of water bodies (Barron
et al., 2012; Scanlon et al., 2023). There are several manifestations of climate change impact on water resources, encompassing
floods, droughts, rising temperatures, deterioration of water quality, and in general intensification of extreme events (Lewsey
et al., 2004; Piccolroaz et al., 2018; Sukanya and Joseph, 2023). These phenomena, combined with anthropogenic alterations
of flows and water quality resulting from various activities (e.g., irrigation, hydropower production, aquaculture), can exert
profound influences on aquatic ecosystems, causing alterations in their structure, function, and overall ecological balance
(Weiskopf et al., 2020; Qu et al., 2020; Antala et al., 2022). Consequently, the establishment of comprehensive monitoring
systems and analytical tools is imperative for accurately quantifying these impacts on aquatic ecosystems.

In the field of river monitoring, technological advancements have significantly improved our ability to assess both water
quantity and quality. Standard monitoring methods for key variables such as water level, temperature, and quality have been
greatly enhanced through the utilization of real-time sensors (Hernandez-Ramirez et al., 2019; Nawar and Altaleb, 2021), the
establishment of cost-effective sensor networks (Meng et al. (2017)), the development of more advanced monitoring instru-
35 ments (Chowdury et al., 2019; Pasika and Gandla, 2020), and the access to remote sensing imagery (Gitelson et al., 1993; Cao
et al., 2021), ultimately leading to heightened precision and reliability. However, it is important to emphasize that none of these
variables provide a direct quantification of the impact of external stressors, whether they arise from natural or anthropogenic
disturbances, on the aquatic ecosystem. Although the use of early-warning indicators based on physical and biological state
variables can be used to predict loss of system resilience and the occurrence of critical transitions, these indicators typically
operate over extended temporal horizons (e.g., decades) and require knowing the underlying mechanisms that steer ecosystem
transitions in order to identify the pertinent state variables (Gsell et al., 2016). When the objective is to assess impacts of ex-
ternal disturbances at the event-time-scale, achieving a good level of evaluation still requires labor-intensive *in situ* biological
sampling with repeated sampling before and after the event (e.g., Metcalfe et al., 2013; Folegot et al., 2021).

A noteworthy source of inspiration can be found in the field of water pollution monitoring, where biotic communities have
45 been used as direct ecosystem indicators since a long time (Cairns et al., 1979; Coker, 1989; Butterworth et al., 2001; Gerhardt
et al., 2006; Li et al., 2010; Holt and Miller, 2011; Siddig et al., 2016). In particular, mussels have been used in biomonitoring
since the mid-1970s with the establishment of the "Mussel Watch" program (Goldberg, 1975) and since then they have been
widely used worldwide as bioaccumulators for the assessment of aquatic pollution (Schöne and Krause Jr, 2016). Going
beyond their employment as bioaccumulators, dating from the 1980's mussels started being explored as potential biosensors
for biological early-warning systems (BEWS) (see e.g., Bae and Park, 2014) for real-time surface and drinking water pollution
monitoring (Guterres et al., 2020; Dvoretsky and Dvoretsky, 2023; Vereycken and Aldridge, 2023). In fact, over 40 years of





studies show that the observation and analysis of mussels' behaviour is a reliable tool for water quality monitoring (Sow et al., 2011), owing to the fact that they change their valve opening and closing activity when they perceive a change in environmental conditions, such as toxicants concentrations (Salanki, 1976; Kramer et al., 1989; Tran et al., 2003, 2007; Beggel and Geist,

2015; Hartmann et al., 2016), food quantity and quality (Higgins, 1980), tidal cycles, and salinity (Davenport, 1981, 1979; Akberali and Davenport, 1982). The immediacy of behavioral responses and the development of simple and cost-effective valve measurement (valvometry) methods have stimulated the production of commercial valvometric systems, such as the Mossel Monitor (Kramer et al., 1989) or the Dreissena Monitor (Borcherding, 1992). The interest in using valvometric responses as an alarm signal directly in real conditions stimulated technological innovations, such as online data systems with remote control

(Sow et al., 2011) and, more recently, applications of artificial intelligence to signal interpretation (Swapna et al., 2022)).

The extensive and successful use of mussels as reliable "biological sensors" for real-time detection of water quality related disturbances, suggests that mussels' valvometry can be a suitable technique also for the automated assessment of the effects of physical stresses, such as the occurrence of floods and droughts or the anthropogenic alteration of flow patterns, on the aquatic ecosystem. These and further hydrological perturbations are increasing in frequency and intensity due to climate change. The

65 extension of mussels' valvometry beyond its initial use in ecotoxicological monitoring of water quality can indeed become important and unique. Recent laboratory tests (Modesto et al., 2023; Termini et al., 2023) were performed on different freshwater mussels' (FMs) populations to investigate the variation of mussels' valve gaping (i.e., the act of partially opening their shells for respiration, filter-feeding, and moving) under different flow discharge and sediment transport scenarios. Valve gaping frequency and opening amplitude (%) was used to analyze mussels' behaviour, according to behavior classifications such as

the one proposed by Hartmann et al. (2016), subsequently extended based on those laboratory tests. Two distinct kinds of behaviour were be identified in non-stressed mussels: normal activity and resting. Regular valve movements related to feeding and moving characterizes normal behavioral activities. Valves constantly opened for filtration/respiration characterize the resting behaviour. Three types of behaviour characterized the mussels' response to stress: transition, adaptation, and avoidance. Transition behaviour can be identified by rapid cycles of abduction (valve-opening) and adduction (valve-closing). The grad-

ual reduction in gaping frequency/amplitude after the transition period can be interpreted as adaptation, i.e. the reduction of responsiveness to ambient stimulation levels through the adjustment of sensitivity. Avoidance behaviour was identified by the closure of valves for a fixed period of time. The results suggested that FMs can be used as BEWS for assessing the impacts of flow discharge variation on riverine biotic communities, paving the way for their application in natural river settings.

The present study, conducted within the framework of the Enterprising PRIN Project (2019), funded by the Ministry of Edu-

80 cation, University and Research (MIUR) of Italy, aims at exploring the use of mussels as an effective real-time BEWS in rivers, with a specific focus on assessing the response of aquatic communities to natural floods. In this regard, this work marks the next phase following the aforementioned laboratory tests. It addresses both the technical challenges linked to the installation of live organisms in the field and the interpretation of the data obtained within the complexity of real-world conditions. The transition from laboratory controlled conditions to the field represents one of the biggest challenges in the development of monitoring

methodologies and protocols. First, the installation of a monitoring system to assess the effects of discharge dynamics on FMs′ behaviour necessitated securing the mussels using cages and/or anchoring systems to prevent them from being displaced by



the flow. Secondly, to prevent the packing of FMs against the downstream wall of the cage during high discharge, we deemed it advisable to secure the FMs to steel rods that are anchored *in situ* rather than allowing them to move freely in the substrate as done in the laboratory tests (Modesto et al., 2023; Termini et al., 2023). The use of steel rods to anchor the FMs was required

also considering that the river bottom of the installation site was characterized by bedrock, which is not an ideal substrate for FMs. The need to block the mussels in an unnatural position, as is commonly practiced to monitor the quality of the aquatic environments (Kramer et al., 1989; Nagai et al., 2006; Robson et al., 2009), may alter the behavioural responses compared to those measured in the laboratory where mussels can freely move within the substrate. This is aspect should be taken into careful consideration when analyzing the results.

With the final aim of proposing the operational use of FMs as a real-time BEWS for hydrological disturbances in rivers, in this study we address three main challenges: i) to define a robust signal processing methodology to analyze the valvometry data and assess the FMs' behaviour, ii) to compare the behavior of free to move and stuck FMs in the laboratory in presence of discharge perturbations, iii) to transfer the experience acquired from laboratory-controlled experiments to applications in real river conditions.

The manuscript is structured as follows. Section 2 provides an overview of the site location and the area where the FMs were collected. It also describes the laboratory and field installation, signal recording and analysis approaches, and the classification of the mussels' behavior. Section 3 presents the results of laboratory experiments and field monitoring. Finally, in Section 4 we discuss the results of the work and draw our final conclusions.

## 2   Materials and method

### 2.1   Field site and mussels' collection

The riverine monitoring site is located along the Paglia River (Italy). The Paglia River (Figure 1) originates in the southeastern region of Tuscany, specifically from Mt. Amiata (1738 meters above sea level). It is located in the central part of Italy and is one of the primary right-side tributaries of the Tiber River. The Paglia River has a length of $86\,\mathrm{km}$ and its basin covers an area of approximately $1320\,\mathrm{km}^2$. The monitoring system based on FMs has been installed in the Paglia River at Orvieto city,

and precisely under the Adunata Bridge, at the right bank of the river where the riverbed is rocky. In addition to this facility, a gauging station was available at this site for monitoring water level and discharge.

A preliminary survey of the river revealed that the native species of the area, *Unio mancus* (Lamarck, 1819) is locally extirpated. Therefore, specimens of the same species collected from the neighboring Lake Montepulciano, Siena Province, Tuscany, Italy (Figure 1) were used for the installation of the valvometry monitoring system.

### 2.2   Data measurement and data collection

In order to monitor the frequency and intensity of FMs gaping, different valvometry methods have been proposed since over one century (reviewed in Vereycken and Aldridge, 2023). In his pioneering work, Marceau (1909) first used a kymograph



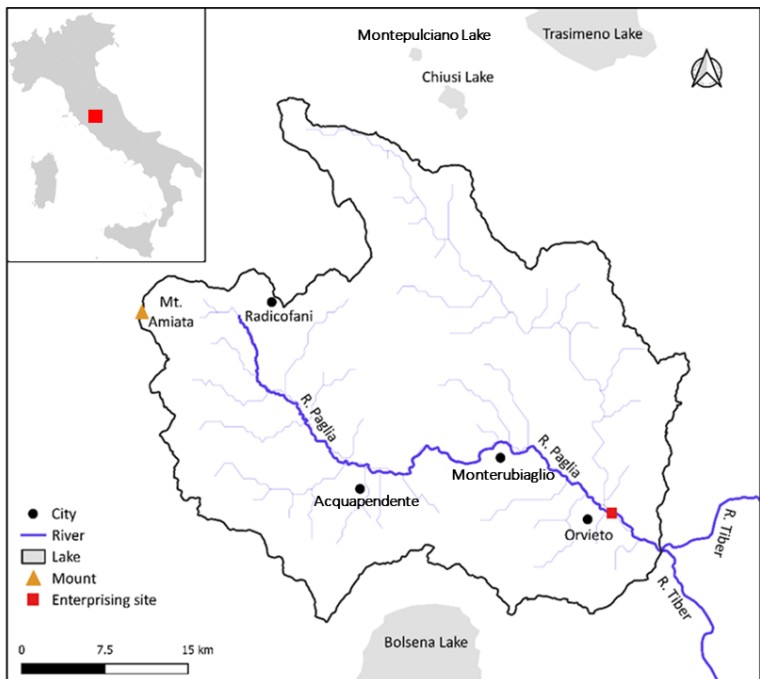

**Figure 1.** Map of the Paglia River and its catchment, showing the location of the field site and of Lake Montepulciano, where the FMs were collected.

to track the valve movement of mussels by attaching a balanced arm equipped with a scribe to one valve of the mussel. Electromagnetic induction to measure the valve displacement was firstly used by Schuring and Geense (1972) and then further developed thanks to technological advancements. Wilson et al. (2002) introduced the use of the Hall effect to record the valve movement of mussels. This approach requires installing a magnet on one valve and a Hall effect sensor on the other valve. The Hall effect sensor measures the magnetic field between the magnet and the sensor itself, which changes according to the distance between the two valves. In this way, both the frequency and intensity of valve gaping can be measured. When the mussel is closed, the magnetic field around the sensor is at its maximum, and when the mussel is fully opened the magnetic field strength around the sensor decreases due to the increased distance between the magnet and the sensor.

In this study, a Hall sensor (Honewy, well SS495A1, 13x10.5 mm, 1.1 g weight) was glued on one side of the mussels' shell, a magnet ($12 \times 10$ mm, $1.8$ g weight) on the opposite side of the shell (Figure 2d). An Arduino board (Mega 2560) was used to record the response of the Hall effect sensor in $\mathrm{mV}$, and then by knowing the minimum and maximum values, the output was normalized and turned to percentage opening (see Section 2.4). An SD card connected to the Arduino was used to store the voltage values. In laboratory experiments, each mussel provided data at a frequency of 1 Hz, while in the field due to a different set-up of the recording system a frequency of 2 Hz was used.




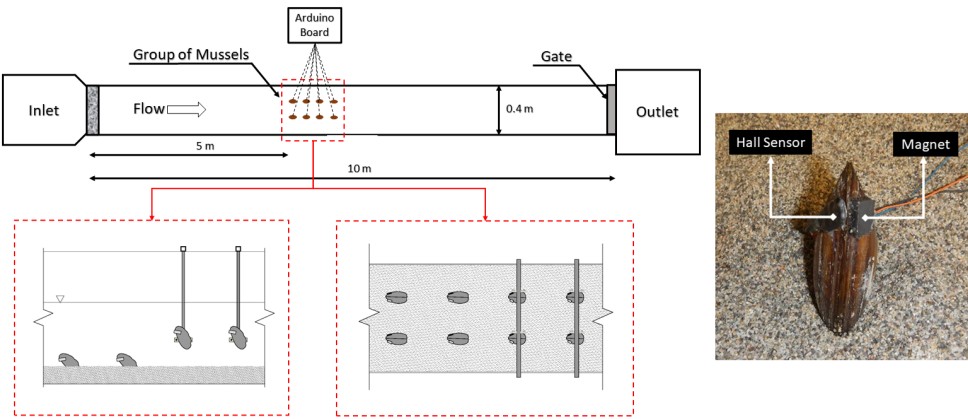

**Figure 2.** a) Experimental setup in the laboratory; b) and c) side and plan views of mussels' arrangement in the flume; d) an example of FM equipped with a Hall sensor and a magnet.

## 2.3 Laboratory experiments and *in situ* installation

FMs have already been experimentally tested as a possible BEWS for flood events using free individuals by Modesto et al. (2023) and Termini et al. (2023) in a laboratory setting. The next step to laboratory tests, i.e., *in situ* application, poses a

number of challenges ranging from the selection of the monitoring site to the choice of the most suitable systems for FMs installation. The exposure of the animals to the parameters to be monitored is one of the main operational challenges in natural environments. While for the monitoring of chemical contamination it is possible to install the animals in lateral derivations of the watercourse or water mains, to monitor the responses to hydrological stresses (i.e., water velocity and turbulence) it is essential to expose the animals in the main channel of the watercourse. This requires installing the FMs in structures that are

as transparent as possible to the flow, ensuring they do not substantially affect the natural flow pattern, and sufficiently stable to guarantee the integrity of both the FMs and the installation throughout the exposure (e.g., Kramer and Foekema, 2001; Sow et al., 2011). For the sake of logistical convenience, the FMs are commonly fixed to solid structures such as steel rods, thus limiting their ability to move. Furthermore, it is important to note that FMs are frequently used in environmental conditions that may differ from their natural habitat, such as when using FMs accustomed to lentic waters in river environments (e.g., Martel

et al., 2003), as is the case in the present study. Consequently, it is crucial to conduct an indepth assessment of FM behavior in these non-native conditions before employing them for monitoring purposes.

     In the context of installing FMs in a large river like the Paglia River, which experiences a broad spectrum of flow conditions (including variations in water level and discharge, hence velocity and turbulence), it was necessary to stick the FMs to rods. To



assess the extent to which limiting their movement affects their behavioral response to environmental stress, an experimental comparison was carried out, analyzing valve movements in both freely moving and immobilized animals. These experiments were conducted before deployment in the field study site, at the Hydraulic Laboratory of the University of Trento (Italy). The experiment was designed to compare the behavior of four free and four stuck FMs belonging to the species *Unio mancus* (Lamarck, 1819) as those used in the field pilot site. The FMs were exposed to the same external conditions for 24 hours (Table 1 , Figure 2). FMs from 1 to 4 were free to move, while the others i.e., from 5 to 8 , were stuck on vertical rods bars by gluing one valve to the rod, that was hung vertically from the top of a $10 \mathrm{~m}$ long flume (Figure 2b). Free and stuck mussels were positioned in the middle of the flume, sufficiently far from the upward and downward boundary conditions. After 10 hours of continuous discharge at a constant rate of $5.3 \mathrm{l/s}$ with $10 \mathrm{~cm}$ substratum (and without sediment transport), the discharge was instantaneously increased to $22 \mathrm{l/s}$ (with bedload and suspended load transport) and maintained at this high value for 2 hours, before returning to the initial baseline value $(5.3 \mathrm{l/s})$. This baseline discharge has been kept constant during the remaining of the experiment. i.e., for the following 12 hours. A further information we looked for is the discharge referring to the incipient condition for sediment transport. We found that $Q = 11~1/s$ is the critical value for the initiation of bedload of the finest grains $(\sim 0.06 \mathrm{~mm})$ forming the sandy sediment mixture of the mobile bed in the flume.

In the pilot site installation in the Paglia River, thirteen FMs were fixed to vertical bars of the same type of those tested in the laboratory and installed in a cage secured at the riverbank. The use of steel rods was necessary to mitigate the risk of FMs being displaced during flood events and due to the unsuitable bedrock substrate at the installation site. The cage was necessary to prevent damages to FMs and electronics. The cage has been designed to ensure robustness while minimizing interaction with the flow. To achieve this, a thin steel frame covered by a coarse grid was used. The Arduino system was installed on the bridge above the riverbank where the cage was positioned. An overview of the installation is provided in Figure 3.

In the location of the FMs cage, a multiparametric probe was installed to measure the water level, temperature, and conductivity. The multiparametric probe is OTT PLS-C that measures water level (resolution: $0.01 \% \mathrm{FS}$; accuracy: $\leq \pm 0.05 \% \mathrm{FS}$ ), temperature (resolution: $0.1^\circ \mathrm{C}$; accuracy: $\pm 0.1^\circ \mathrm{C}$ ), and conductivity (resolution: $1 \mu \mathrm{S/cm}$; accuracy: $\pm 1 \mu \mathrm{S/cm}$ ).

**Table 1.** Laboratory Experiments were performed.

| Type of Specimen | Total number of mussels | Number of free mussels | Number of stuck mussels | Duration (hrs) | Low Discharge (l/s) | High Discharge (l/s) | Variation Discharges (l/s) |
|---|---|---|---|---|---|---|---|
| Unio mancus | 8 | 4 | 4 | 24 | 5.3 | 22 | 16.7 |

## 2.4 Signal processing

The raw valvometry data measured using the Hall effect sensor were expressed in $\mathrm{mV}$ and were indicative of the distance between the two valves. Describing the behavior of FMs in terms of the frequency and intensity of gaping using raw data may not be straightforward due to inherent physiological variations among FMs, primarily influenced by their size and shape, as well as the nonuniform attachment of magnets and sensors to the individual FMs. For this reason, in order to have a common



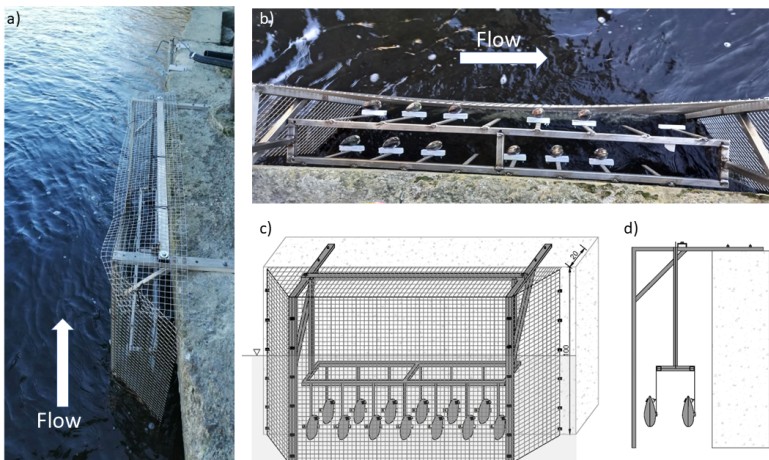

**Figure 3.** Field Installation - Enterprising pilot site, Orvieto City, Italy: a) flow direction; b) top view; c) front view; d) side view.

frame of response among all mussels, the opening signals were normalized between 0 to $100\%$ : $0\%$ indicates that the mussel's valves are fully closed, and $100\%$ that the mussel's valves are fully open. Before normalizing the signal, possible outliers due to occasional acquisition artifacts have been removed. In this context, outliers have been defined using the 0.1 and 99.9

percentiles as the lower and upper threshold bounds, respectively. The removed points were subsequently reconstructed through interpolation, and the signal was finally normalized based on the minimum and maximum values, thereby standardizing all FMs signal within the same reference frame ranging from 0 to 1 :

$$\hat{x}_i(t) = \frac{\max(x_i) - x_i(t)}{\max(x_i) - \min(x_i)} \tag{1}$$

where $x_i(t)$ is the value of the raw signal (in mV) of FM $i$ at time $t$ and $\hat{x}_i(t)$ is the corresponding normalized opening value.

We recall that $x_i(t)$ decreases with the distance between the two valves, while the dimensionless variable $\hat{x}_i(t)$ is proportional to the opening. By multiplying $\hat{x}_i(t)$ times 100, the corresponding percentage opening ranging from $0\%$ to $100\%$ is obtained. As a side note, it is worth emphasizing that to effectively normalize a signal according to equation 1, it is imperative for the signal duration to be sufficiently long to encompass both the periods when the FM is fully closed and fully opened.

The resulting FMs signals were analyzed with the aim of identifying the occurrence of change points in the FMs behaviour.

As discussed in the Introduction, these changes may be linked to the normal behaviour of non-stressed FMs, but also to the response of these organisms to external perturbations. The monitoring of a sufficiently large number of FMs allowed us to discriminate between the specific behaviour of individual FMs driven by their own activity, and a systematic response of the FMs community to external disturbances. Abrupt change points in the mean of the opening signals were identified using Matlab *findchangepts* function.



Parallel to abrupt changes of behaviour characterized by stepwise discontinuities in the opening signal, it has been observed that when FMs are subject to stress they exhibit marked changes in both the frequency and intensity of their gaping (Modesto et al., 2023; Termini et al., 2023). The statistical analysis of the FMs gaping frequencies was carried out using Continuous Wavelet Transform (CWT), a mathematical technique that decomposes a signal into different frequency components. CWT is particularly useful when dealing with non-stationary signals. Indeed, unlike traditional Fourier analysis, CWT can capture both high and low-frequency variations in time-series data, making it especially effective for analyzing signals that exhibit dynamic changes in frequency and amplitude over time (Meyers et al., 1993; Rhif et al., 2019).

The CWT analysis is based on the convolution of a signal $f(t)$ with a set of functions $\psi_{ab}(t)$, known as wavelets, derived from translations and dilations of a so-called mother wavelet $\psi(t)$ :

$$\psi_{ab}(t) = \frac{\psi}{\sqrt{a}}\left(\frac{t-b}{a}\right) \quad a,b \in R, a > 0 \tag{2}$$

where $a$ is known as the scale factor and $b$ defines a shift in time. Different mother wavelets can be used to decompose a signal, all of which must meet specific conditions (see e.g., Meyers et al., 1993). The convolution of the signal $f(t)$ with set of wavelets is the wavelet transform:

$$T_{\psi}(a,b) = \frac{1}{\sqrt{a}}\int_{-\infty}^{+\infty}\psi^{*}\left(\frac{t-b}{a}\right)f(t)dt \tag{3}$$

where $*$ denotes the complex conjugate and $T_{\psi}(a,b)$ is the wavelet coefficient (which, for the sake of completeness, depends not only on $a$ and $b$ but also on the choice of the mother wavelet $\psi$ ). In this way, the signal $f(t)$ is analyzed by comparing it to a set of wavelet functions $\psi_{ab}$ characterized by continuously varying scale $a$ and shift $b$. Unlike sinusoidal functions in Fourier analysis, these wavelets functions do not have a fixed frequency. Rather, they are versatile mathematical functions inherently flexible in both time and frequency domains, which adapt to the non-stationary characteristics of the signal being analyzed. The scale factor $a$ is inversely related to frequencies: smaller scales correspond to more "compressed" wavelets thus higher pseudo-frequencies, and capture details in the signal at shorter time scales, while larger scales correspond to more "stretched" wavelets thus lower pseudo-frequencies, capturing broader features at longer time scales. Note that the term pseudo-frequencies is often used to emphasize that these values should not be confused with the fixed frequencies associated with sinusoidal waves.

The results of the CWT analysis can be effectively visualized through the use of scalograms and pseudo-frequency (or scale)-averaged wavelet spectra. The scalogram is a graphical representation of signal power distribution across various pseudo-frequencies and through time. It is constructed by considering the magnitude of the complex wavelet coefficients and allows for a comprehensive examination of how different pseudo-frequencies and times contribute to the overall power of the signal. The pseudo-frequency-averaged wavelet spectrum provides a summary of the signal's energy distribution across multiple scales, offering insights into both localized and broad-frequency features present in the signal over time. The pseudo-frequency-averaged wavelet spectrum is obtained by scale-averaging the magnitude-squared scalogram over all scales.

In this study, CWT has been computed by applying the Matlab *cwt* function using the Morse wavelet as the mother wavelet to the time series signal of each FMs, after detrending and removal of abrupt changes in the mean of the opening signal.



Detrending was required to remove possible low frequency trends, while the identification and removal of step changes in the mean was needed to avoid introducing artifacts in the results. In fact, the CWT decomposition of a signal featuring an abrupt step change will result into a combination of high-frequency components, which encapsulate the abrupt transition, along with
230 lower-frequency components of the signal, delineating the signal's smoother and gradual variations, across the entire spectrum of frequencies. This would generate an artifact in the resulting scalograms and pseudo-frequency-averaged wavelet spectra, possibly hindering the interpretation of the informative features of the signal. All step changes in the mean of the opening signals were therefore removed and the signal detrended before the CWT analysis.

In order to obtain a synthetic summary of the results, the scalograms obtained from the wavelet analysis of all FMs have
235 been combined into one, corresponding to the median scalogram. Similar to the signal pre-processing described above, in order to get a consistent frame of reference, the scalogram of each FM has been normalized between minimum and maximum values after removal of outliers. This allows us to effectively appreciate the existence of consistent features across FMs and characterize them in terms of dominant pseudo-frequencies and position in time. The summary pseudo-frequency-averaged wavelet spectra has been obtained from the median scalogram.

## 3 Results

### 3.1 Laboratory results

The median opening signal of the FMs measured during the 24-hour laboratory experiment along with the 25th and 75th percentiles are shown in Figure 4a, for free and stuck mussels separately. The evolution of discharge in time is also shown in the second axis. The figure clearly shows that both groups of FMs responded to the discharge increase occurred 10 hours
after the start of the experiment with a sharp and localized change in the median opening (thick colored lines). By examining the shaded area of this plot (representing the 25th and 75th percentiles), it becomes evident that the signals from free FMs exhibit a more complex and various behavior compared to those from stuck FMs. The former group displays a larger number of features, most of which are not directly associated with hydrodynamic changes. This is further evidenced in Figure S1 in the Supplementary Material, which illustrates the distinctive behavioral patterns exhibited by each FMs and the major
discontinuities in the mean of the signal. The difference between the two groups of mussels can be attributed to the limited mobility of stuck FMs compared to free FMs, whereas the restriction of behaviors like walking and drifting leads to a simpler signal for stuck mussels. Notably, all stuck FMs responded by closing their valves when the discharge increased, while two out of the four free FMs responded with an increase of the opening (FMs numbered 1 and 2) and one with a decrease (FM number 4). Apart from exhibiting some noise, likely arising from electrical issues, which, however, did not significantly impact the
results, free FM numbered 3 displayed a behavior similar to that of FM numbered 4 (refer to Figure S1). While both free and stuck FMs clearly and promptly responded to the rapid increase in discharge, as evidenced by the change in the mean valve opening discussed above, only stuck mussels displayed a similar response upon the reestablishment of the base flow (12 hours after the start of the experiment), albeit to a lesser degree (Figure 4a and Figure S1). Moving beyond the analysis of mean valve opening, however, a distinct signature of the discharge perturbation is discernible in both free and stuck FMs through the





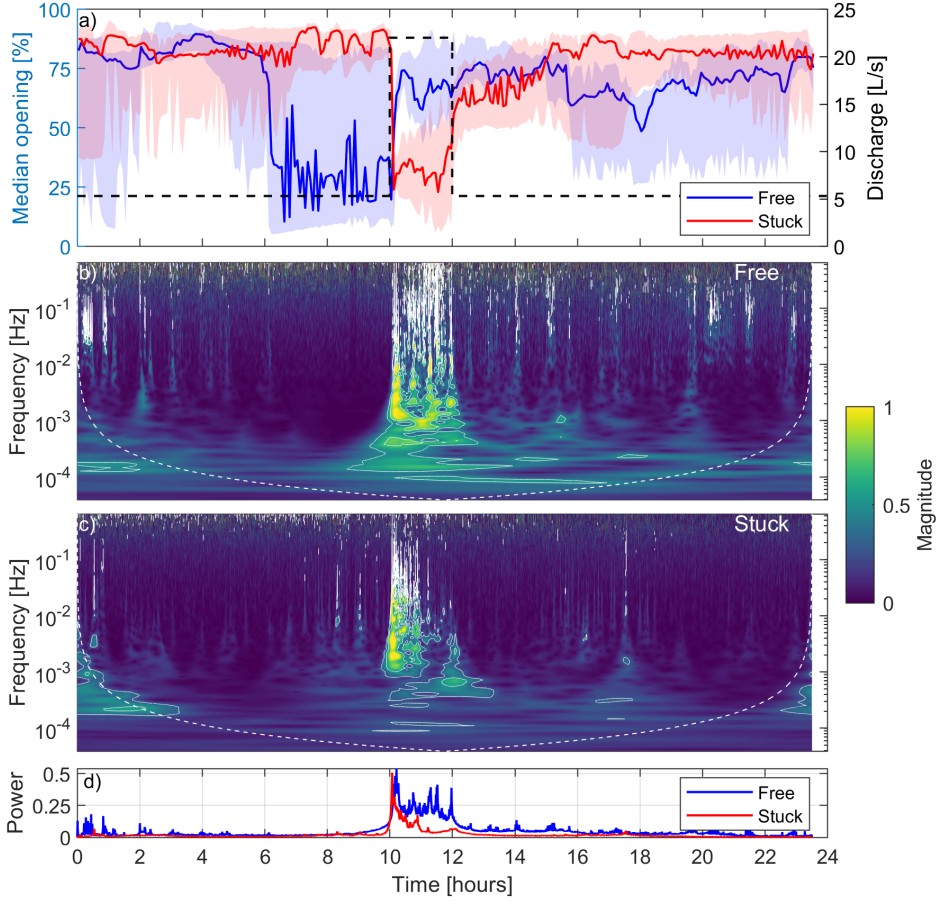

**Figure 4.** Laboratory experiment: a) left y-axis: median valve opening signals of free and stuck mussels with 25th and 75th percentiles indicated by the shaded area; right y-axis: discharge; b) median scalogram of the free mussels; c) median scalogram of the stuck mussels; d) pseudo-frequency-averaged wavelet spectrum. White contours in b) and c) represent the 95th and 99th percentiles of the CWT coefficient.

occurrence of broad-frequency features localized over time around the perturbation period (see the period between 10 and 12 hours in Figure S1). Such features are clearly appreciable looking at the scalograms of free (Figure 4b) and stuck (Figure 4c) FMs. Prior to the increase in discharge (from the beginning of the experiment to 10 hours), the FMs gaping was characterized by low pseudo-frequencies (below $10^{-3}$ Hz ) with less energy. However, following the increase in discharge, they prompted began responding across the whole range of pseudo-frequency range (up to 1 Hz ), displaying higher energy levels (as indicated by the more yellowish regions). The white contours on the scalogram plot represent the 95th and 99th percentiles of the CWT coefficient and were used to emphasize the energy-rich areas. The response is similar between free and stuck FMs, although stuck FMs adapted quicker to the new discharge conditions compared to free FMs. This is evident also looking at Figure 4d that shows the pseudofrequency-averaged wavelet spectrum for both FMs' classes: stuck mussels' power returned to normal



after an hour after the discharge increase, followed by free mussels who responded with high power until the end of the event
(two hours after the discharge increase). In both cases, the dominant pseudofrequencies after the discharge have been re-set to
values consistent with those characterizing the first part of the experiment.

## 3.2 Field results

Because laboratory experiments showed overall consistent responses between free and stuck FMs in the presence of hydro-
dynamic stresses, this supported the possibility of installing stuck FMs in the field. Thirteen stuck FMs were installed in the
afternoon of March 30, 2022 at the Enterprising pilot site along the Paglia River (see Figures 1 and 3). Figure 5a illustrates
the signals of the thirteen stuck FMs from 10 PM on March 30, 2022 until midnight on April 1, 2022. It should be mentioned
that there is a gap of an hour and a half in the data for all FMs, with missing signals from mussels between 10.5 and 12 hours
since the start of the signal (i.e., between 8:30 AM and 10:00 AM on March 31) due to unspecified technical issues. According
to this plot, all FMs except for mussels numbered 2, 3, and 12 experienced a significant shift in the mean valve opening with
a generalized closing of the valves approximately 5.5 hours after the start of the time series, specifically around 3:30 AM on
March 31, coinciding with a flood event in the river. As can be observed in the data, the sensor installed on mussel numbered
2 experienced technical issues, preventing its use in the analysis. On the other side, the sensor installed on FM numbered 3
and 12 were operating normally but the FMs were already closed before the flood event (likely because they were in the state
of resting, see Introduction), hence not displaying any additional closure but a minor and progressive opening and gaping.
All the other ten FMs were characterized by normal behavior before the flood event, with their valves open and characterized
by regular valve movements as expected during respiration and filtration. These ten mussels, which were exhibiting normal
behavior before the flood, displayed avoidance behavior by closing their valves immediately upon the onset of the flood and
changing their gaping frequency.

In general, in terms of change in the mean valve opening, the mussels responded similarly to changes in hydrodynamic
conditions as observed in the flume experiment, as illustrated in Figure 5b, which depicts the median valve opening signal of
the FMs (all individuals except for FM numbered 2) in relation to the changes in water level measured by the multiparametric
sensor (see also Figure S2). According to the water level line, the median valve opening reveals two main discontinuities. The
first discontinuity is evident at 5.5 hours from the start of the time series (i.e., at 3:30 AM), when the water level, as measured
by the multiparametric sensor, rapidly increased from $0.3 \, \mathrm{m}$ (corresponding to $4 \, \mathrm{m^3/s}$) to $2 \, \mathrm{m}$ (corresponding to $140 \, \mathrm{m^3/s}$).
This rise in water level marked the onset of the flood event, which was characterized by a sharply rising front. This discontinuity
in the valve opening signal is clearly visible when examining both the individual FM signals (Figure 5a) and the median signal
(Figure 5b). The second discontinuity clearly emerges only by looking at the median signal (Figure 5b), while it is hindered
in the individual time series. It occurred at about 23 hours after the start of the time series (i.e., at 9PM), when the water level
raised from $0.6 \, \mathrm{m} \, (18 \, \mathrm{m^3/s})$ to $1 \, \mathrm{m} \, (45 \, \mathrm{m^3/s})$, thus interrupting the previously gradual decrease in water level following the
initial peak.

The FMs' response to the hydrodynamic disturbance, as reflected in changes in the frequency of gaping, is shown in the
scalogram of Figure 5c. This plot is obtained excluding FM numbered 2, which was affected by the technical issues discussed



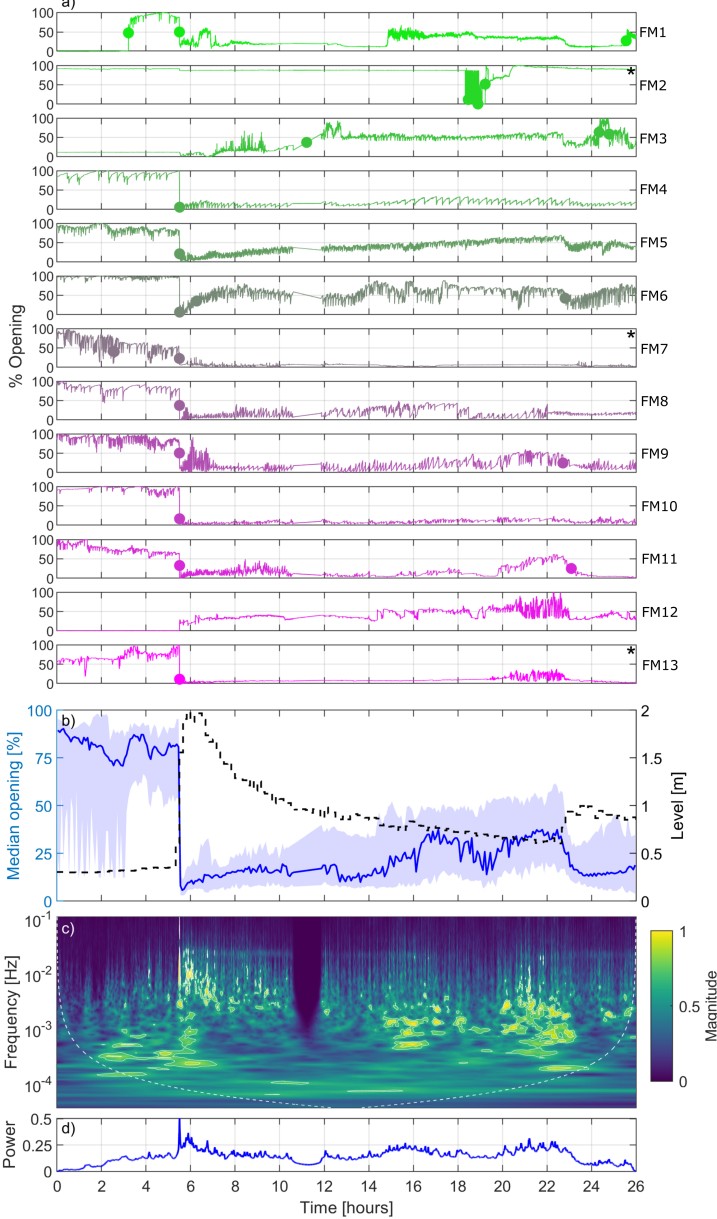

**Figure 5.** Results from the thirteen FMs deployed at the Paglia River pilot site during the flood on March 31, 2022; a) valve opening signals for the individual FMs (dots indicate abrupt change points in the mean of the opening signals when the mean opening changes by more than 25%; * depicts FMs that are excluded from the wavelet transform analysis); b) left y-axis: median valve opening signals with 25th and 75th percentiles indicated by the shaded area; right y-axis: water level; c) median scalogram of the FMs; d) pseudo-frequency-averaged wavelet spectrum. White contours in c) represent the 95th and 99th percentiles of the CWT coefficient.





above, and for the sake of clarity also FMs numbered 7 and 13, respectively characterized by intense activity before the flood and a significantly damped response after the flood, to an extent that was not in alignment with the behavior of the other

FMs. These mussels are labeled with * in Figure 5a. As indicated by the scalogram, the mussels exhibited responses at a low frequency (below $10^{-4} - 10^{-3}$ Hz ) before the peak of the flood, displaying signals of lower energy. As soon as the level raised, parallel to the sudden closure of the valves seen in Figures 5a and b, the FMs showed a generalized response shifted towards higher frequencies (up to 0.1 Hz ) and characterized by higher energy levels (yellowish regions). This response gradually diminished as the discharge decreased, resulting in lower frequencies and reduced energy levels. Between 15 to 23 hours from

the beginning of the time series, the water level, and consequently the discharge, transitioned into the latter phase of the flood event, characterized by minor fluctuations over time. Additionally, the temperature returned to higher values and stabilized (see Figure S2). During this time window, the FMs underwent a slight opening of the valves (Figure 5b) and intensified their gaping around frequencies of $10^{-3}$ Hz (Figure 5c) trying to restore their normal activity. Following the occurrence of the second, smaller peak in water level (approximately 23 hours after the start of the time series), all FMs closed their valve once more

(Figure 5b) and the majority of them exhibited a significant decrease in the frequency of gaping (Figure 5a and c). The overall picture is summarized also in the pseudo-frequency-averaged wavelet spectrum shown in Figure 5d, which clearly indicates the instantaneous evident response of the FMs to the main perturbation, similar to what observed in the laboratory experiments.

## 4   Discussion and conclusion

In recent research, based on laboratory experiments Modesto et al. (2023) and Termini et al. (2023) proposed FMs as effective

BEWS for assessing the impact of hydrodynamic stresses on the aquatic ecosystem. In this study, for the first time, FMs were deployed and evaluated in real river conditions, in the Paglia River at Orvieto, Italy. The initial challenge we faced was securing the mussels in place and preventing them from being carried away by the river's current. In consideration of other *in situ* exposure methods (e.g., Kramer and Foekema, 2001; Sow et al., 2011), we opted to glue the mussels to vertical rods anchored at the riverbank. To ensure that the use of stuck mussels did not significantly alter their behavioral responses, we

conducted controlled laboratory experiments to confirm that stuck and free mussels exhibit consistent reactions under the same hydrodynamic stressors.

The results demonstrated that stuck mussels not only produce consistent signals when compared to free mussels but also exhibit less complexity, primarily due to the reduced number of features resulting from movement constraints (Figure 4). Consequently, these signals can be more easily interpreted and associated with perturbations in external conditions. In fact, stuck

mussels have limited means of response, primarily relaying on gaping, as they are unable to engage in actions such as escaping or burying themselves more deeply. It is worth noting that concerning mussel behavior classification, stuck mussels demonstrated a faster adaptation in response to a prolonged stimulus, facilitating the faster restoration of pre-event gaping frequencies. This could be explained by the positive correlation between the intensity of the stimulus and the degree of adaptation (Capraro et al., 1979; Hollins et al., 1990). In fact, stuck mussels experience a stronger stimulus that free mussels due to the impossibility

of actively searching for shelter at the event time, but the results show they have a shorter adaptation period. Overall, based



on the laboratory comparison, we could confidently assert that the installation of stuck mussels in real river conditions ensures both site stability during floods and the representativeness of the results.

Field data acquired at the pilot site in the Paglia River provided a solid validation of the effectiveness and reliability of using stuck mussels as part of a real-time BEWS. All the FMs exhibited a synchronized and highly distinct response to a flood
event occurred during the monitoring campaign, promptly transitioning their behavior in terms of both mean valve opening and gaping frequency. Of paramount importance for practical applications is the observation that these mussels responded immediately as the discharge increased, effectively detecting the flood at its very onset, these results confirm and validate what has been observed in the laboratory. The sharp and rapid mussel reactions can be explained by the almost abrupt increase of levels and discharges of the flood, and also by the values of discharges around the peak, which, as in the flume experiments
(see Section 2.3), triggered some sediment transport. To establish for the Paglia River an incipient (alias critical) discharge value, we exploited the shear-stress distribution relative to the flood on May 31, 2022, determined by a 3D Reynolds-Averaged Navier-Stokes (3D-RANS) model described in Bahmanpouri et al. (2023), and some bed material samples we collected from the bed surface just upstream and downstream of the Adunata Bridge site. With a constant flow discharge corresponding to that observed at the flood peak, the shear stress distribution reached a maximum of about $100\,\mathrm{Pa}$ under the bridge, and smaller
values away from the bridge. Looking at the numerical results, a reference value of $30\,\mathrm{Pa}$ is exerted over a large part of the flow domain. From the bed material sample, we found that the $d_90$ (i.e., diameter corresponding to the $90^{\mathrm{th}}$ percentile of the granulometric distribution) of the sediments forming the substrate of the bed is about $0.01$ m. With these values, the well-known dimensionless Shields stress number $\theta = \tau/\left[(\rho_s - \rho_w)\,gd\right]$, where $\rho_s\,(\rho_w)$ is the sediment (water) density, $g$ the gravitational constant, and $d$ the characteristic particle diameter, during the occurrence of the flow, reached a value of about $0.2[-]$, well
above the critical one, which can be assumed equal to about 0.06, as in many references (see e.g., Pähtz and Durán, 2018). Indeed, according to the 3D-RANS model, a critical condition for the Paglia River at the Adunata Bridge corresponds to about $30\,\mathrm{m}^3/\mathrm{s}$. The ratio, in terms of peak flood-discharge over the critical value, is therefore about (or greater) than 5 , whereas in the flume experiments the mussels experienced a discharge about double the critical one, as from the data reported in Section 2.3. Our laboratory and field results show that the reactivity of the mussels is efficient in a wide range of peak/critical discharge
ratios. However, the utilization of FMs in real riverine conditions did present logistical challenges. While the monitoring station at the Paglia River was operational for several months, the data acquired only pertains to the occurrence of a flood. The lack of significant events during the monitoring period and the mortality of some FMs prevented the acquisition of additional data that would have been valuable for the analysis. Additionally, during the prosecution of the monitoring activity after the flood event occurred in March 30, 2022, it happened that metal particles occasionally suspended in the water, likely originating from
an upstream mine, accumulated on the magnets installed to the FMs, thus altering the valvometric signal. Efforts to expand the dataset for a more comprehensive analysis are envisaged and undergoing.

To improve the interpretation of behavioural signals, we proposed a statistical analysis based on the combined identification of abrupt change points in the mean of the opening signals and application of continuous wavelet transform to the detrended time series after removal of these discontinuities (refer to Figures S3 and S4 to view the cleaned signals). This approach has
proven to be a robust and reliable tool for FMs' signal processing, able to effectively identify the animals' response to external



stressors. This identification can be achieved both visually, through the examination of scalogram plots, and quantitatively, through the generation of associated pseudo-frequency-averaged wavelet spectrum plots. For the sake of comparability, the scalogram of each FM has been normalized between minimum and maximum values after the removal of outliers. Then, the median scalogram has heen calculated and used to identify dominant pseudo-frequencies and their temporal positions. This approach enables comparisons within the same frame of reference between different FMs and between median scalograms obtained under various conditions (e.g., in the laboratory and in the river). Similarly, the pseudo-frequency-averaged wavelet spectra derived from the median scalograms can be fairly compared to discern shared patterns and distinctions across the various setups. Notably, results from both the laboratory and the field indicate that during external stress events (i.e., an increase in discharge), the power of the spectrum reached and exceeded a value of approximately 0.25. While further data collection from additional sites is needed before proposing a threshold value as a simple indicator of aquatic ecosystem stress, these findings underscore the effectiveness of combining FMs valvometry and CWT processing towards the establishment of real-time operational BEWS.

The results obtained pave the way for the utilization of this analysis framework in operational BEWS employing stuck FMs. The main conclusions of the present study can be synthesized as follows:

- both free and stuck freshwater mussels can serve as effective ecosystem warning indicators in aquatic environments, with the choice between them depending on the riverbed and flow rate conditions;

- sticking the mussels to supports constrains their behavior, but this results in sharper event detection compared to free mussels and easier interpretation of the signals; item[-] continuous wavelet transform proves to be a valuable tool for interpreting the FMs signals, in terms of identifying pseudo-frequency features present in the signal over time and using them to describe the response of FMs to external perturbations;

- laboratory and field experiments with stuck mussels demonstrate their response to hydrodynamic stresses within a frequency of valve gaping ranging from $10^{-3}$ Hz to 1 Hz. This frequency range is larger than the background frequency range during normal behavior (around $10^{-4} - 10^{-3}$ Hz when taking the median across multiple individuals). These frequency values correspond to conditions that indicate the presence of stressful conditions for the FMs, thus underscoring the potential use of FMs as real-time BEWS for identifying potential threats to the aquatic ecosystem;

- the comparison between pseudo-frequency-averaged wavelet spectra obtained in the laboratory and in the field suggests the potential introduction (subject to further data acquisition) of a simple indicator based on power values to detect disturbances in the aquatic environment.

*Data availability.* The data and code that support the study are available from the corresponding author upon request.



*Author contributions.* **Ashkan Pilbala:** Investigation, Methodology, Formal Analysis, Data Curation, Writing - Original Draft. **Nicoletta Riccardi:** Conceptualization, Investigation, Methodology, Resources, Supervision, Writing - Original Draft. **Nina Benistati:** Investigation, Data Curation, Writing - Review & Editing. **Vanessa Modesto:** Investigation, Resources, Writing - Review & Editing. **Donatella Termini:** Investigation, Data Curation, Writing - Review & Editing. **Dario Manca:** Data Curation. **Augusto Benigni:** Data Curation. **Cristiano Corradini:** Data Curation. **Tommaso Lazzarin:** Data Curation, Writing - Review & Editing. **Tommaso Moramarco:** Conceptualization,
Project administration, Funding acquisition, Writing - Review & Editing. **Luigi Fraccarollo:** Conceptualization, Investigation, Methodology, Supervision, Writing - review & editing. **Sebastiano Piccolroaz:** Conceptualization, Investigation, Methodology, Formal Analysis, Data Curation, Visualization, Supervision, Writing Original Draft.

*Competing interests.* All authors declare they have no financial interests.

*Acknowledgements.* This work has been supported by the Italian PRIN 2017 project Enterprising (2017SEB7Z8).



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
