# Peer review of "Figure S1. Valve opening signals for the individual free and stuck FMs deployed in the laboratory experiment (dots indicate abrupt change points in the mean of the opening signals when the mean opening changes by more than 25% )."

_EGUsphere, 2023_

## Author Comment (AC2)

Dear Prof. Cynthia Maan,

We would like to thank you for taking the time to review our manuscript and provide valuable comments. We are delighted that our work has attracted your attention. We greatly appreciate the insights you have provided, which will undoubtedly enhance the quality of our research. In the following pages, we reply (in black) to each of the comments (*in blue*).

*C1. Figure 4: Why does the median opening of the free mussels suddenly decrease at 6 hours after the start, without stimulus? Why is the median opening of the stuck mussels increasing clearly before the end of the discharge peak, and why is the median opening of the free mussels not reacting to the end of the stimulus? The response of the frequency is more straight forward/ convincing.*

A1. The fact that the median opening shows a decrease after 6 h is not linked to a systematic decrease across the four free FMs. Indeed, Figure S1, which reports the signal of all mussels separately, shows that before the stimulus the signals from the free FMs are different, according to their independent activities. This is also evident in Figure 4, where the 25th-75th percentile confidence bound is wide before the stimulus but shrinks as the discharge increases, indicating a coherent response across the four free FMs. Generally, we will emphasize in the revised manuscript that the use of the opening alone limits understanding of the mussel's behavior. Instead, our analysis clearly shows the value of analyzing the frequency response for a deeper interpretation and use of the signal. In the revised manuscript, we will also emphasize the reference to Figure S1, which will facilitate the reading of the median behavior and avoid confusion. In addition, we note that the frequency response is more evident because the amplitude is only variable within the range determined by the length of the adductor muscle. This varies with mussel size and, to a lesser extent, with intraspecific variation in shell shape and muscle insertion location. Therefore, amplitude is considered a secondary parameter concerning frequency variations, except in cases where the animal is completely closed (avoidance) or moribund (muscle relaxation before death).

*C2. Figure 5: could you indicate the time-span of the missing data (10-12h) in the figure? The response to the stimulus in this experiment seems different than the response in the laboratory experiment in the way that there is no clear 'recovery' back to the original values. Based on the laboratory experiment, would you not expect a faster recovery (compared with the measurements in the river) of the (stuck) community?*

A2. We will highlight in the Figure the missing data from 10 to 12 h caused by some technical issue. Thank you for the suggestion.

As for the recovery to pre-stimulus animal conditions, an in depth analysis of this specific aspect needs further verification. Contrary to laboratory conditions, in the field the environmental conditions may not completely return to the pre-stimulus state. The recession stage of a flood, by instance, is quite longer than the physiological recovery time the mussels showed in the laboratory runs. This involves other aspects, such as the skill of

mussels to resist prolonged external variations (see also our reply A8, below). Furthermore, in order to extrapolate laboratory results to natural conditions, it is necessary to consider the greater variability of boundary conditions with respect to the parameter being examined. 1) The response of the mussels is an expression of a reaction to changes in multiple conditions that can only be controlled and restored to previous conditions in the laboratory. 2) Mussels that are immobilised undergo an unnatural constriction that alters their ability to return to pre-stress conditions. For these reasons, further field experiments should be conducted in order to investigate the method's limitations and to develop protocols for addressing them adequately. Also, we note that in the field high discharge conditions lasted for a long time after the initial peak and it is not excluded that within the time window analyzed here, the FMs were constantly stressed by hydrodynamic conditions. We will add specific comments to the Discussion section of the revised manuscript.

*C3. Line 10: The stuck mussels produce signals that are 'more consistent' with flood occurrence. I wonder if this is true because for the free mussels the frequency is increased over the full period of enhanced stress, whereas the frequency of the stuck mussels falls back to lower frequencies well before the period of enhanced discharge/ stimuli (figure 4). Also, such consistency would be beneficial if the aim was to measure hydrodynamic conditions. However, the impact on biotic communities might be overestimated due to the ' larger consistency'. If the link between the stimuli/ stress factors and the free mussels (so the 'reality') is weaker (?), can the response of the stuck mussels still be an indicator for free-community behavior or stress?*

A3. The term "consistent" was misused or misinterpreted here. What we mean here is that signals from immobilized FMs show a higher level of consistency between each other, rather than with the occurrence of floods. We adjusted the sentence as follows: "Moreover, immobilised mussels produced more interpretable signals than free-moving mussels due to the reduced number of features resulting from movement constraints." The key point is that the signals from immobilised mussels are easier to interpret since they are cleaner (fewer features due to restricted movement). In addition, our laboratory experiments demonstrated that immobilised mussels (required in the field for logistical reasons, as commented in the manuscript) can be used to detect when the community is stressed by external stimuli, similar to the use of free mussels. In this case, the aim is to introduce a real-time biological early warning system (BEWS). An alarm system needs to detect a fault in a timely manner (see also A5) and overestimation is always preferable to underestimation. The precautionary approach commonly applied to monitoring and defining toxicity thresholds is adopted precisely to avoid underestimating the risk. According to the same principle, to make real-time warning systems more effective, the most sensitive species that are or could be present in the target environment are used.

*C4. Line 250-251 : is "whereas" the right word to use in this sentence?*

A4. Thank you for pointing this out, in the revised version we corrected the sentence as follows:

"The difference between the two groups of mussels can be attributed to the limited mobility of stuck FMs compared to free FMs. This limited mobility caused by restricting behaviors like walking and drifting leads to a simpler signal for stuck mussels."

*C5. Line 273 and line 327: the reactions of stuck and free mussels are not that "consistent": the glued mussels didn't maintain the high frequency over the full period of enhanced stress (falls back to the original frequency before the period of enhanced discharge ends), and there is a difference between the reactions in terms of opening: the opening amplitude of free mussels increased whereas the amplitude of stuck mussels decreased (figure 4).*

A5. The aim of this study is to introduce a real-time biological early warning system (BEWS). To this end, firstly, the onset time of changes is critical and then, as further information, the type/duration of response of the mussels to the stimulus can be helpful. The BEWS, based on the wavelet signal processing presented here, has been shown to be successful in identifying when the FMs are undergoing a stress response, when either free or immobilised FMs are considered. In this respect, we can state that the responses of the two types of FMs are consistent (see also Figure 4, and in particular subplot d). Further information, such as the duration of the behaviours, gives the reader a better visualisation and is of interest for deep biological features, but is not the main goal here. In this regard, two other papers published by our group have dealt with the behaviour of mussels to stimuli in the laboratory, focusing on behaviours after the onset of the stimulus, such as adaptation and avoidance (Modesto et al., 2023; Termini et al., 2023).

*C6. Line 311: "additionally the temperature .." how does the temperature fit in this story? is it relevant? Could there be an impact of the temperature on the FMs frequency and opening amplitude and/or community ?*

A6. FMs can change their gaping behavior in response to fluctuations in environmental conditions such as water depth, light, temperature, and particulate matter (Tran et al., 2003; Ropert-Coudert & Wilson, 2004; Robson et al., 2009). We conducted laboratory experiments which showed that for the species/population used the frequencies increased significantly above 28°C. The variation is much stronger if the temperature variation is rapid because the animal is unable to adapt. The experiments are still under development and the results will be published soon. For the interpretation of the results included in this publication, the temperature should not be influential because it is always below the tolerance threshold of the eurythermal generalist species that we used. We will stress this in the revised manuscript and adjust possible misleading sentences accordingly.

*C7. Line 313: "trying to restore their.." They are not nearly close to their original opening amplitude or frequency.*

A7. We agree and have removed the comment. As noted in A2, the response of mussels in the field reflects a response to changes in multiple conditions that are difficult to control in real riverine systems. The interpretation of this signal is therefore complex and

requires further investigation. However, what is clear is the response at the onset of flooding, which supports the use of FMs and the processing approach to establish a real-time BEWS as discussed in the manuscript.

*C8. Line 332 ".. faster adaptation in response to a prolonged stimulus" Can this really be seen as 'adaptation'? Or do the stuck mussels get tired sooner? i.e. would it be beneficial for them to return faster to the lower default frequencies, even when the discharge is still enlarged?*

A8. Both hypotheses are probably true: we have verified that there is a tendency towards adaptation in both free and stuck mussels, but it is logical to think that the latter gets tired sooner. All these aspects, little studied in the laboratory and not yet studied in the field, must be studied in depth for the development of an applicable methodology. It is for this reason that we decided to carry out this first attempt at validation in the field in order to clarify aspects that have been neglected until now, although systems based on the use of mussels as alarm sentinels are already marketed (e.g. mosselmonitor). It is worth mentioning here that more data will be needed to fully understand the FMs' behavior in the field and, accordingly, we plan to acquire more data in the future.

**References**

Modesto, V., Tosato, L., Pilbala, A., Benistati, N., Fraccarollo, L., Termini, D., Manca, D., Moramarco, T., Sousa, R., and Riccardi, N.: Mussel behaviour as a tool to measure the impact of hydrodynamic stressors, Hydrobiologia, 850, 807–820, ISSN 1573-5117, 2023. https://doi.org/10.1007/s10750-022-05126-x

Robson, A. A., G. R. Thomas, C. Garcia de Leaniz & R. P. Wilson: Valve gape and exhalant pumping in bivalves: optimization of measurement. Aquatic Biology 6: 191–200, 2009. https://doi.org/10.3354/ab00128

Ropert-Coudert, Y. & R. P. Wilson: Subjectivity in biologging science: do logged data mislead? Memoirs of National Institute of Polar Research 58: 23–33, 2004. https://doi.org/10.1016/j.anbehav.2003.09.010

Termini, D., Benistati, N., Tosato, L., Pilbala, A., Modesto, V., Fraccarollo, L., Manca, D., Moramarco, T., and Riccardi, N: Identification of hydrodynamic changes in rivers by means of freshwater mussels' behavioural response: an experimental investigation, Ecohydrology, p. e2544, ISSN 1936-0584, 2023. https://doi.org/10.1002/eco.2544

Tran, D., P. Ciret, A. Ciutat, G. Durrieu & J. C. Massabuau: Estimation of potential and limits of bivalve closure response to detect contaminants: application to cadmium.

Environmental Toxicology and Chemistry 22(4): 914–920,2003. https://doi.org/10.1002/etc.5620220432

---

## Author Response (AR1)

Prof. Thom Bogaard

Editor-in-Chief

Hydrology and Earth System Sciences (HESS)

26 Feb 2024

Dear Prof. Thom Bogaard

Please find the revised version of the manuscript entitled "Real-Time Biological Early Warning System based on Freshwater Mussels' Valvometry Data" Ref. No.: EGUSPHERE-2023-2405. We appreciate the comments provided by one anonymous reviewer and their insightful and valuable suggestions. In our opinion, the quality of the manuscript is improved after revising the manuscript according to the reviewers' comments. The responses to the reviewers' comments are appended at the end of this letter.

Thank you very much for your consideration.

Yours Truly,

Corresponding author

Dr. Sebastiano Piccolroaz

Department of Civil, Environmental, and Mechanical Engineering

University of Trento, Trento, Italy

via Mesiano, 77, 38123,

Email corresponding author: s.piccolroaz@unitn.it

Dear Reviewer #1,

we thank you for the positive assessment of our manuscript and for providing constructive comments and useful suggestions that we have included in the revised version of the manuscript. In the following pages, we respond (in black) to each of the Reviewer's comments (in *blue*). Please notice that pages and lines in this document refer to the original manuscript present in open discussion.

*C1. In the Abstract, I think that you could mention what was done in a more logical order, as it was the laboratory experiments that helped inform what was best to do in the following field experiment – i.e. include the sentence "The experimental results demonstrate that stuck mussels produce signals…" before "Subsequently, we examined the response of…".*

**A1**. We thank the Reviewer for the suggestion. The abstract has been updated accordingly, and in general it has been rewritten in order to clarify the background and objectives of the study. The new abstract is provided below:

*Quantifying the effects of external climatic and anthropogenic stressors on aquatic ecosystems is an important task for scientific purposes and management progress in the field of water resources. In this study, we propose an innovative use of biotic communities as real-time indicators, which offers a promising solution for directly quantifying the impact of these external stressors on the aquatic ecosystem health. Specifically, we investigated the influence of natural river floods on riverine biotic communities using freshwater mussels (FMs) as reliable biosensors.*

*Using the valvometry technique, we monitored the valve gaping of FMs and analysed both amplitude and frequency.*

*The valve movement of the FMs was tracked by installing a magnet on one valve and a Hall effect sensor on the other valve. The magnetic field between the magnet and the sensor was recorded using an Arduino board, and its changes over time were normalised to give the percentage opening of the FMs. The recorded data was then analysed using the Continuous Wavelet Transform (CWT) analysis to study the time-dependent frequency of the signals. The experiments were carried out both in a laboratory flume and in the River Paglia (Italy). The laboratory experiments were conducted with FMs in two configurations: freely moving on the bed and immobilised on vertical rods. Testing of the immobilised configuration was necessary because the same configuration was used in the field in order to prevent FMs from packing against the downstream wall of the protection cage during floods or from breaking their connection wires. These experiments allowed us to verify that immobilised mussels show similar responses to abrupt changes in flow conditions as free mussels. Moreover, immobilised mussels produced more neat and interpretable signals than free-moving mussels due to the reduced number of features resulting from movement constraints. We then analysed the response of thirteen immobilised mussels in real river conditions during a flood on 31 March 2022. The FMs in the field showed a rapid and significant change in valve gap frequency as the flood escalated, confirming the general behaviour observed in the*

*laboratory results in the presence of an abrupt increase in the flow. These results highlight the effectiveness of using FMs as biosensors for the timely detection of environmental stressors related to natural floods and emphasise the utility of CWT as a powerful signal-processing tool for the analysis of valvometry data. The study proposes the integration of FM valvometry and CWT for the development of operational real-time Biological Early Warning Systems (BEWS) with the aim of monitoring and protecting aquatic ecosystems. Future research should focus on extending the investigation of the responsiveness of freshwater mussels to specific stressors (e.g. turbidity, temperature, chemicals) and on testing the applications of the proposed BEWS to quantify the impact of both natural stressors (e.g. heat waves, droughts) and anthropogenic stressors (e.g. hydropeaking, reservoir flushing, chemical contamination).*

**C2.** *I would also recommend changing the word "stuck" to "immobilized" throughout the manuscript, as the former sounds too colloquial.*

**A2.** Thank you for pointing this out. We have changed the word "stuck" to "immobilised" throughout the manuscript.

**C3.** *It seems that the Materials and Method section could be written more concisely. Make sure that when you first refer to the field site, you also then call it the "Enterprising site" (as it appears in Fig. 1), and consistently use a single term for the site throughout, as it seems strange to switch between multiple terms for the same thing later in the manuscript. You could also perhaps give a grid reference of the site. It may also be sensible to use a different shade for Italy in Fig. 1, as it seems too similar in shade to that of the Lakes.*

**A3.** The field site is now referred to as "Field monitoring site" throughout the manuscript. The Figure has been updated as suggested by the Reviewer. We revised the Materials and Methods section in order to make it clearer, more concise and to the point.

**C4.** *The letters seem to be missing from Fig. 2. I recommend you label each animal as it is numbered – i.e. say "FM4 did…" rather than "the FM numbered 4 did…". As you effectively used two cohorts of FMs, 8 in the laboratory and 13 in the field, I would recommend each having its own number or letter – perhaps call animals in laboratory FMa to FMh and animals in field FM1 to FM13.*

**A4.** We thank the Reviewer for this suggestion. We have referred to the animals in the text as FMx (rather than FM number x) and labeled the animals in the figures. Also, we clarified that the animals used in both cases were of the same species, so as not to confuse the reader.

**C5.** *I think there is no need to include Table 1, as these details are clear enough in the text, and are not so extensive as to necessitate a table.*

**A5.** We agree with the Reviewer's opinion and we removed Table 1.

*C6. Make sure you refer to the discharge in L/s, with a space between the number and unit – i.e. at L157, 158, 159, 161.*

**A6**. We thank the Reviewer for detecting these typos. In the new version of the manuscript, all of them were corrected.

*C7. It would probably be best to consistently use the term, "vertical rods", rather than changing to "bars" occasionally.*

**A7.** We agree with the suggestion and we corrected it in the revised version.

*C8. When mentioning the probe, I think it is unnecessary to provide the resolution and accuracy.*

**A8.** We agree with the suggestion. Since this information is not fundamental here, we removed it from the revised version of the manuscript.

*C9. I find L173-188 needlessly long. I think the important details could easily be said more succinctly. I also think it is unnecessary to include the equation for what is essentially only converting values to percentage opening using the minimum and maximum voltage readings. I think it is important to make clear that the scale was assumed to be linear.*

**A9.** According to the Reviewer's comment, we removed Eq. 1 and shortened L173-188.

*C10. Could you provide a little more detail on the Matlab function/mean (L194) – I presume it is a running mean calculated within an individual over a certain time frame?*

**A10.** We modified the sentence as follows:

*Abrupt change points in the mean of the opening signals were identified using the Matlab function \textit{findchangepts}, an iterative procedure that detects significant transitions in time-series data through adaptive segmentation of the original time series.*

*C11. Perhaps you could label where the multiparametric probe is positioned in Fig. 3.*

**A11**. We agree with the Reviewer and added a label for the position of the multiparametric probe in Fig. 3.

*C12. I also wonder whether you could include a little more detail and justification for detrending and removal of step changes, otherwise it could sound arbitrary.*

**A12**. To prevent the introduction of artefacts into the results, as detailed in L227-232, the removal of step changes was deemed necessary. In fact, when decomposed step changes are characterized by a mix of high-frequency and lower-frequency components in the signal, which are spurious and may obscure the real signal characteristics. Likewise, detrending, involving the removal of mean

and linear trend, is a standard signal processing procedure employed to eliminate low-frequency components without affecting the high-frequency content of the original signal (which is the important component of the signal here). The paragraph at L225-233 has been revised for enhanced clarity.

*In this study, CWT was computed by applying the Matlab \textit{cwt} function using the Morse wavelet as the mother wavelet to the time series signal of each FM, after removal of abrupt changes in the mean of the opening signal. Identifying and removing step changes in the mean of the signal was necessary to avoid introducing spurious results. In fact, when a CWT decomposition is performed on a signal with an abrupt step change, the result is a mixture of high-frequency components that capture the abrupt transition and lower-frequency components that describe the smoother and more gradual changes in the signal, across the entire frequency spectrum. The presence of abrupt changes would generate an artefact in the resulting scalograms and pseudo-frequency-averaged wavelet spectra, possibly hindering the interpretation of the informative features of the signal. Step change removal was achieved by detrending the segments of the signal between two successive step changes (identified as discussed above), i.e. by subtracting the mean and removing the linear trend, hence without altering the informative, high-frequency content of the original signal.*

**C13.** *I think you should include key dates, such as when you collected specimens, when you attached sensors, how long after attachment of sensors that you started work, as it is important to know the time frames. I also think you should include temperatures the specimens were maintained at for completeness.*

**A13.** We added this sentence to the Materials and method: *A preliminary survey of the river revealed that the native species of the area, \textit{Unio mancus} \citep{Lamarck_1819} is locally extirpated. Therefore, specimens of the same species were collected from the neighbouring Lake Montepulciano, Siena Province, Tuscany, Italy (Figure \ref{fig:F1}) on March 29, 2022, and they were maintained in a tank filled with lake water. The mussels were divided into two groups, a group was installed at the field monitoring site in the afternoon of March 30, 2022, while the other group was sent to the Hydraulics Laboratory of the University of Trento (Italy) for the flume experiments. On arrival at the laboratory, the animals were acclimated for two weeks in a $500$ L recirculating flow-through aquarium with aerated water and gravel-sand substrate, and fed with a mixed culture of natural algae. Details of the laboratory and field installation are given in Section \ref{s:description_exp}.*

**C14.** *I think the Materials and Method section should be slightly restructured to provide details in a more logical way and avoid repetition as far as possible. I think you can start as you have with*

*the site choice, how that dictated choice of mussel species for investigating both in the laboratory and in the field. Then give the details of the Hall sensor method. Then I think you should give the details of the conversion of the mV readings, then highlight that you were particularly interested in the opening amplitude – that you extracted the median opening for multiple specimens and looked at the mean within each specimen, with Matlab determining abrupt changes – and the frequency, using CWT, including an explanation of that. Then describe the laboratory work, explain its purpose was to determine if there were differences between free and immobilized mussels because only immobilized mussels could realistically be used in the field. Then give details of what the work in the river entailed. I think you need to make it clear that you deliberately recorded behavior in the field during a flood event here too*.

**A14.** According to the Reviewer's suggestion we re-organized the Materials and Methods section introducing the case study and the collection of the FMs, presenting the laboratory and in-situ installation, describing the valvometry data collection and, finally, describing the signal processing technique. Also, we removed unnecessary repetitions.

*C15. L250 "The difference…" – this sentence explains the observation, which is a point that should really be saved for the Discussion section.*

**A15**. We agree that the sentence highlighted by the Reviewer is an interpretation of the results that could fit into the Discussion. We prefer to leave the sentence where it is, as it helps to understand the results. However, we recalled and expanded the concept in the Discussion.

*C16. Make sure you consistently use either "pseudo-frequency" or "pseudofrequency" but not both.*

**A16.** Corrected in the revised version.

*C17. In Fig.s 4 and 5, it would be best to have a little more detail than just "Magnitude" for the scalograms.*

**A17.** We revised the captions of the figures specifying that these plots refer to "*scalogram showing the median normalised magnitude of the continuous wavelet transform over the…*". Also, in the main text we specified that "*[scalograms are] constructed by considering the absolute value (or magnitude) of the complex wavelet coefficients*", hence explaining what we mean by "magnitude".

*C18. I think in the Results section, the observation that the immobilized laboratory mussels returned to baseline behavior quicker should be highlighted but inferring that this suggests they adapted quicker (and what this suggests) should be kept for the Discussion.*

**A18**. Similarly to above, we prefer to keep this sentence in the Results as it helps to understand what is shown in the figure. However, we recalled and expanded the concept in the Discussion.

*C19. L273 – I wonder if you should say consistent "responsiveness" rather than "responses", as the actual response of free and immobilized were only similar through CWT analysis (though of differing duration) and rather different in terms of opening amplitude.*

**A19.** We agree with the Reviewer's recommendation.

*C20. Also, I recommend saying it "supported the suitability" rather than "possibility" (L274). I think the dates should be mentioned in the Materials and Method section rather than here, although it is obviously fine to mention the missing data/technical issue here.*

**A20.** Thank you, we corrected the sentence. Also, we reported the relevant dates and timing (e.g., the installation date) in the Materials and Method section.

*C21. Also, probably best to refer to the shift as "marked" rather than "significant" unless significance tests were done.*

**A21.** The changes were detected using the *findchangepts* procedure described above, which required setting a threshold to identify change points. We therefore agree that the term "marked" is better, as no significance tests are performed by default.

*C22. I would avoid classifying the behavior and explaining the observations ("state of resting", "regular valve movements as expected during respiration and filtration", "displayed avoidance behavior" and "trying to restore their normal activity") and save explanation and classification of behaviour for the Discussion. I also think that it would be best to slightly change the structure, i.e. to first mention the observations in the hydrodynamic conditions, then mention the behavioral observations, and then to tie the observations within both together, rather than giving details sporadically.*

**A22.** According to the Reviewer's suggestion, we partially moved the comment on the FMs behaviour to the Discussion section. However, we prefer to leave the following sentence in the Results section as it is required to correctly interpret the Results for FM3 and FM12:

"*On the other side, the sensors installed on FM3 and FM12 were operating normally but the FMs were already closed before the flood event (likely because they were in the state of resting, see Introduction), hence not displaying any additional closure but a minor and progressive opening and gaping. All the other ten FMs were characterized by normal behaviour before the flood event, with their valves open and characterized by regular valve movements as expected during respiration and filtration.*"

*C23. The details of which mussel(s) were excluded from Fig. 4 b-d should be mentioned in the caption for that figure.*

**A23**. No FMs were excluded from Figure 4. We have noted this in the caption by saying that the plots represent the results for all free and all immobilised FMs.

*C24. In the Discussion, make sure you highlight all the key findings and explain observations where appropriate – first from the laboratory, then the field. As mentioned before, I think "responsiveness" may be more appropriate than "responses" (in L325). Details of method/calculation of the incipient discharge (L345-359) should be included in the Materials and Method section, and the outcome of the calculation included in the Results, with only its implications mentioned here. I think the details about logistical challenges (L360-366) should be mentioned in brief, and only after the findings. After that I would include a reiteration of the particular value of the findings, specifically how it extends the use of valvometry for hydrodynamic factors (including ideas mentioned in Introduction). Then you can include the bullet points.*

**A24**. We have restructured the Discussion section as suggested by the Reviewer, moving some sentences from the Results to the Discussion, moving the logistical challenges after the presentation of the results, and adding some comments on the possible extensions of the valvometry technique. As for this last point, we added the following text before the bullet points:

*The results obtained pave the way for the utilisation of the valvometry technique and of the signal processing framework presented here in operational BEWS in different contexts. Freshwater mussels can serve as indicators to quantify the impact of both natural stressors (e.g. heat waves, droughts) and anthropogenic stressors (e.g. hydropeaking, reservoir flushing, chemical contamination) on the aquatic ecosystem. As such, they can be instrumental in reporting the impacts of climate change on water resources and in the management and permitting processes implemented by local authorities. Future research should focus on extending the investigation of the responsiveness of freshwater mussels to other stressors (e.g. turbidity, temperature, chemicals) and on verifying the effectiveness of the signal processing technique presented here in identifying possible synthetic indicators related to different stressors.*

As for the incipient discharge evaluation, since this is a corollary comment, we prefer to keep the definition of the well-known formula for the Shields parameter in the Discussion section. However, we anticipated some concepts about sediment transport in the Materials and Method (with reference to the laboratory experiments). Also, we note that we found an inaccuracy in the value of the Shields parameter reported in the text. Although the message does not change, we now report the Shield parameter for the peak discharge (0.6 instead of 0.2). Also, we found that the critical discharge is not 30 m3/s, as reported in the original submission, but about 4 m3/s. As a courtesy to the Reviewer, we have reported here the distribution of the simulated Shields parameter at Q=160 m3/s and Q=4 m3/s.

[Figure]

Figure R1: spatial distribution of the Shields parameter for 160 m3/s and 4 m3/s.

*C25. I think the Supplementary Figures should all have been referred to in the Results section and no figures should really be referred to in the Discussion.*

**A25.** Following the Reviewer's suggestion, we have referenced all figures in the Results section for the first time (some are referenced in the Discussion where appropriate).

*C26. I presume it should read as "d90" in L351, and I presume it should not read "0.2[-]" in L354. I presume that a separate bullet point starts at "continuous wavelet transform proves…" in L388.*

**A26**. We corrected d_90 into d_{90} and removed [-] (meaning dimensionless).

*C27. In the Supplementary Figures, I think the plots should have labels for the FM from which the data was taken (as in Fig. 5a). I presume it would be better for the captions of Fig.s S3 and S4 to refer to the "removal of step changes". I also think it might be better for what appears as Fig. S2 to instead appear after what is currently Fig. S3, and for the supplementary figures to all be mentioned in the Results section.*

**A27**. In Figure S4, we have added the * symbol to FMs that are not considered in the signal processing, as in Figure 5. We have changed the caption of Figures S3 (now Figure S2) and S4 as suggested by the Reviewer. All figures are now mentioned for the first time in the Results section. We updated the order of the figures in the Supporting Information according to the Reviewer's suggestion.

*C28. Make sure that you use a consistent version of English. I noticed that most words used US English spelling, but a few used the British version of "behaviour" and "set-up" too. The manuscript is generally easy to understand. It may be beneficial for a native English speaker to read through it to make minor changes.*

**A28**. Thank you for highlighting this aspect, we revised the manuscript carefully and used only words with UK English spelling.

*C29. Here are some possible changes that would make it easier to read as written English:*

*"FMs" should appear as "FMs'" in L3,5, in L189 (second occurrence), in L262 (second occurrence), and as "FM" in L15,135,169, in L189 (first occurrence), and L193,197,226,249,268*

*"have been" should be changed to "were" in L4,234, and "has been" should be changed to "was" in L109,225,236,239,373,374*

*L12 – change "thus confirming" to "consistent with"*

*L34 – format of source seems to be inconsistent*

*L40 – change "temporal horizons" to "time frames"*

*L42 – little unsure of what you mean by "event-time-scale" (For Sebastiano)*

*L55 – reference order should be 1979 then 1981*

*L60 – remove the extra bracket on the reference*

*L65-66 – change the end of the sentence ("become important and unique") to something that does not suggest its use before was neither of these things, such as by saying "enhance the importance and highlight the unique insight of this approach".*

*L69-70 – change "was" to "were"*

*L71 – remove "be"*

*L76 – change "was" to "is"*

*L93 – remove "is"*

*L95 – change "final" to "overall", so it's clear that this is the main aim of the work, not the last in a list of aims*

*L114 – add comma after "(Figure 1)" and change to "for the laboratory and in situ installation"*

*L126 – presume should read "Honeywell"*

*Make sure that when giving dimensions, you use a consistent format in spacing, i.e. L126 and 127*

*L148 – insert "considered" before "necessary", as it was a conscious decision*

*L164 – change "was necessary" to "was deemed necessary"*

*L244 – remove "occurred" and put a comma before and after "10 hours after the start of the experiment"*

*L247 – change "various" to "varied" and "those from" to "that of"*

*L248 – change "are not" to "appear not to be" as it is speculative*

*L263 – change "prompted" to "promptly"*

*L264 – change "pseudo-frequency" to "pseudo-frequencies"*

**A29.** Thank you for your comments. All points have been addressed.

---

## Author Response (AR2)

Dear Editor,

Please we submit the revised version of our manuscript with the technical corrections requested by the Reviewer: labels in Figure 2 and restructuring of a bullet point in the conclusions.

Thank you for your consideration.

Yours sincerely,

Sebastiano Piccolroaz (on behalf of the co-authors)